# Novel Biomarkers in Evaluating Cardiac Function in Patients on Hemodialysis—A Pilot Prospective Observational Cohort Study

**DOI:** 10.3390/diagnostics14060664

**Published:** 2024-03-21

**Authors:** Lazar Chisavu, Viviana Mihaela Ivan, Adelina Mihaescu, Flavia Chisavu, Oana Schiller, Luciana Marc, Flaviu Bob, Adalbert Schiller

**Affiliations:** 1Centre for Molecular Research in Nephrology and Vascular Disease, Faculty of Medicine ‘Victor Babes’, 300041 Timisoara, Romania; chisavu.lazar@umft.ro (L.C.); ivan.viviana@umft.ro (V.M.I.); chisavu.flavia@umft.ro (F.C.); marc.luciana@umft.ro (L.M.); bob.flaviuraul@umft.ro (F.B.); schiller.adalbert@umft.ro (A.S.); 2Department of Internal Medicine II–Division of Nephrology, “Victor Babeș” University of Medicine and Pharmacy, 300041 Timisoara, Romania; 3Department of Internal Medicine II–Division of Cardiology, “Victor Babeș” University of Medicine and Pharmacy, 300041 Timisoara, Romania; 4Louis Turcanu’ Emergency County Hospital for Children, 300011 Timisoara, Romania; 5AVITUM BBraun Romania, 300041 Timisoara, Romania; oanaleo67@gmail.com

**Keywords:** hemodialysis, mortality, cardiovascular disease, novel biomarkers, cardiac function and structure

## Abstract

Chronic kidney disease patients treated by hemodialysis present a high cardiovascular morbidity and mortality. There is an imperative need for novel biomarkers for identifying these patients and to offer possible therapeutically interventions. We performed a prospective observational cohort study on 77 patients in the period of October 2021–October 2023. We measured serum plasma levels of interleukin 1-beta, galectin 3, human suppression of tumorigenicity factor 2, bone morphogenetic protein 2 and fibroblastic growth factor 23 at the inclusion site. We evaluated the correlations of these biomarkers with cardiac function and structure evaluated by echocardiography. The mean age was 61.02 (±11.81) years, with 45 (56.2%) males and with a dialysis vintage of 4.95 (2.4–7.8) years. Median ejection fraction was 51 (43–54%), and more than two-thirds of the patients presented valvular calcifications. Overall mortality was 22%. Interleukin 1-beta was correlated positively with ejection fraction and global longitudinal strain and negatively with left atrium diameter and left ventricle telesystolic diameter. Galectin 3 values were negatively correlated with aortic valve fibrosis and mitral valve calcifications, and human suppression tumorigenicity factor 2 was negatively correlated with mitral valve calcifications. Some of these novel biomarkers could be used to better assess cardiovascular disease in patients on maintenance hemodialysis.

## 1. Introduction

Chronic kidney disease (CKD) patients present a high cardiovascular disease (CVD) risk and CVD-related mortality [1]. This risk increases with a decrease in kidney function, and CVD mortality becomes the main cause of death in kidney failure patients treated by hemodialysis (HD) [2,3]. Even though more than 50 years have passed since hemodialysis became a reliable method of renal replacement therapy, cardiovascular events remain the leading cause of death among these patients [4].

In recent years, in order to detect and reverse the mechanisms involved in CVD alterations and mortality in CKD patients, biomarkers have been used to explore inflammation, accelerated atherosclerosis, oxidative stress, cardiovascular calcifications associated with mineral bone disorder (CKD-MBD) and so on [5,6,7]. Many of these studies used clinical hard end-points and biological quantifiers to prove and validate the utility, predictability and, at the end, usefulness of the explored biomarkers. Less research has explored the influence of the above-cited mechanisms (i.e., inflammation, CKD-MBD and so on) on cardiac structural and functional alterations, detectable by cardiac ultrasound (known to be associated with CVD mortality in CKD), using biomarkers.

Interleukin 1-beta (IL-1) is a pro-inflammatory cytokine involved in tissue repair, cell growth and inflammatory response [8]. It is a well-known fact that patients on HD have a pro-inflammatory status, and the increased IL-1 level is well documented in CKD and HD patients [9,10,11,12]. Some studies have explored the connections between IL-1 and cardiac structure with promising results [13,14,15,16].

Galectin 3 is a soluble β-galactoside-binding lectin (250 amino acids) that has regulatory roles in cell proliferation, tissue repair, inflammation and fibrogenesis [16]. Several studies have suggested that increased levels of galectin 3 are associated with renal fibrosis [17,18]. In addition, there are studies suggesting that galectin 3 plays a central role in heart failure pathophysiology through myocardial inflammation and fibrosis, thus becoming a predictor of heart failure [19,20]. Some studies evaluated the impact on all-cause mortality and cardiovascular events and showed that higher levels of galectin 3 are an independent risk factor for death in HD patients [21,22,23].

The suppression of tumorigenicity 2 (ST2) gene was discovered in 1989 and is part of the IL-1 gene cluster [24]. The main products of transcription of this gene are ST2L (or IL1RL1-b) which is a membrane receptor for the IL-1 family and sST2a (or IL1RL1-a), which is the soluble receptor that can be identified in plasma [24]. Several articles proved the prognostic utility of sST2a in heart failure and its association with left ventricle hypertrophy (LVH) in CKD, hypertension and metabolic syndrome patients [25,26,27]. In a recent meta-analysis regarding sST2a levels in HD patients, it seems that increased levels are associated with higher all-cause mortality [28].

Bone morphogenetic protein (BMP) is a member of the superfamily of transforming growth factor. The most important isoform is BMP2, which regulates cell growth and differentiation [29]. The role of BMP2 in vascular calcification, atherosclerosis and inflammation is well established, but only one study evaluated the BMP-2 levels in HD patients [30,31].

Fibroblastic growth factor 23 (FGF23) is an osteocytic hormone with effects on calcitriol production and increased phosphate excretion. Furthermore, it appears that FGF23 is an independent risk factor for cardiovascular disease [32]. It seems that besides the bone production of FGF23, the myocytes also produce FGF23 in certain conditions [32]. The impact of FGF23 on the myocardium is still debated, with some studies suggesting a negative impact of elevated FGF23 and others suggesting a positive one [32].

In the face of these scarce data on specific biomarkers and their implications for CVD in HD patients, we evaluated all the aforementioned biomarkers and their relationships with cardiac structure and function in HD patients. In addition, we explored the impact on mortality of these biomarkers.

## 2. Material and Methods

This observational prospective cohort study was conducted between October 2021 and October 2023. All the patients signed an informed consent form upon inclusion, and the Ethics Committee of the Dialysis Centre approved the study (Nr. 33/30 June 2021). The study respected the declaration of Helsinki regarding human studies and was in accordance with the Ethics Code of the World Medical Association. The inclusion criteria in this study were, besides the informed consent of the patients, a history of at least 3 months on stable hemodialysis.

The parameters were retrieved on the same day as the dialysis session, before the beginning of hemodialysis, after the longest weekly period between dialysis sessions (for patients who perform dialysis on days of Monday, Wednesday and Friday, the data were recorded on Monday, and for patients who perform dialysis on Tuesday, Thursday and Saturday, the data were retrieved on Tuesday). All the patients performed three dialysis sessions per week of at least 4 h in duration, using high-flux dialyzers.

We noted the presence of arterial hypertension after the evaluation of patients’ recorded diagnostics from HD files.

The cardiac assessment was performed during the second and third hours of dialysis (Pulse Doppler, M-mode continuous and two-dimensional) on Monday or Tuesday, respectively. A single cardiologist using the same echocardiography device performed the measurements in order to limit the bias. Echocardiography was performed in concordance with European Association of Cardiovascular Imaging (EACI) recommendations. The left ventricular ejection fraction (LVEF) was assessed using the Simpson method, and we noted heart valve calcifications and fibrosis, interventricular septum (IVS), left ventricle telediastolic diameter (LVTDD), right ventricle diameter (RV), global longitudinal strain (GLS), endomyocardial calcifications, aortic atheroma, left ventricle telesystolic diameter (LVTSD), left ventricle mass (LVM), left atrium diameter (LA) and E/A rapport.

The samples for special biomarkers were collected as per the manual instructions for each biomarker.

Human bone morphogenetic protein 2 (BMP2) was quantified using an ELISA kit with the catalog no: E-EL-H0011 and product size: 96T/48T/24T/96T*5 from Elabscience. The sensitivity of the kit is 37.5 pg/mL, with a detection range of 62.5–4000 pg/mL.

Fibroblastic growth factor 23 (FGF-23) was quantified using an ELISA kit with the catalog no: E-EL-H1161 and product size: 96T/48T/24T/96T*5 from Elabscience. The sensitivity of the kit is 9.38 pg/mL with a detection range of 15.63–1000 pg/mL.

Interleukin 1-beta (IL-1B) was quantified using an ELISA kit with the catalog no: E-EL-H0149 and product size: 96T/48T/24T/96T*5 from Elabscience. The sensitivity of the kit is 4.69 pg/mL, with a detection range of 7.81–500 pg/mL.

All the samples were performed using the patients’ plasma. Plasma was collected using EDTA-Na2 as an anticoagulant. After the samples were centrifuged for 15 min at 1000× *g* at a temperature ranging from 2 to 8 degrees Celsius in the first 30 min after collection, the supernatant was collected for the assay.

BMP2, FGF23 and IL-1B analysis was performed at the laboratory of the Clinical Emergency County Hospital in Timisoara, Romania, using the aforementioned kits.

Galectin 3 and suppression of tumorigenicity 2 (ST2) analysis was performed in an external laboratory. The blood samples for these two biomarkers were collected in vials with EDTA-Na2 anticoagulant.

## 3. Statistical Analysis

Data are presented as average ± standard deviation (SD) for numerical variables, with Gaussian distribution, median and interquartile range (IQR) for numerical variables and non-Gaussian distributions and percentage from the sub-group total and number of individuals for categorical ones. The Shapiro–Wilk test was performed for normality assessment for continuous variable distributions and Levene’s test for equality of variances. Multivariable regression and logistic regression models assessed the individual impact of several confounding factors. A repeated backward-stepwise algorithm was performed in order to find the most appropriate theoretical model (exclusion criteria *p* > 0.1 and inclusion criteria *p* < 0.05). Kaplan–Meyer survival curves were assessed to evaluate the impact on mortality. In addition, we performed Cox regression hazard curves to assess the impact of cofounding factors on mortality. A *p* value < 0.05 was considered for statistical significance. The software package used for statistical analysis was MedCalc^®^ Statistical Software version 22.009 (MedCalc Software Ltd., Ostend, Belgium; https://www.medcalc.org (accessed on 14 February 2024); 2023).

## 4. Results

The baseline characteristics of the 77 patients included in the study are presented in Table 1. The mean age of the cohort was 61.02 (SD of 11.81 years), with a median dialysis history of 4.95 years (IQR: 2.4–7.8 years) and male majority (56.2%). The nutritional status was good, with serum albumin at 4.1 (IQR: 3.9–4.3 g/dL), predialysis creatinine at 8.54 (SD 2.05 mg/dL) and reduced inflammation with a C-reactive protein level of 0.43 (IQR: 0.26–1.2 mg/dL). The patients presented an in-target hemoglobin level (10.85 with IQR: 10.3–11.6 g/dL), in-target mineral bone disease management with the calcium phosphorus product below 55 mg^2^/dL^2^ (47.86 with SD of 13.06 mg^2^/dL^2^) and parathyroid hormone between two and nine times the maximum normal value (428 with IQR 197–713.5 ng/mL). The patients were not acidotic (serum bicarbonate of 22 with a SD of 2.27 mmol/L) and presented light hyperkalemia (5.37 with SD of 0.64 mmol/L).

Cardiac parameters for all the patients are presented in Table 2. Overall, they presented a slightly increased left atrium diameter (41.87 with SD of 4.7 mm), 76.9% with aortic atheromatosis, a high incidence of calcification (endomyocardial—70.5%, aortic valve—67.9%, mitral valve—78.2%), important valvular fibrosis incidence (aortic—80.8%. mitral—78.2%), a normal ejection fraction (51% with IQR of 43–54%) and increased interventricular septum (13 IQR 12–14 mm) and left ventricle mass (254.5, IQR 203–304 g). Arterial hypertension was present in 92.2% of the cohort (71 out of 77 patients).

The values of specific biomarkers are presented in Table 3. FGF-23 values are near the upper normal limit (47.93 with IQR 21.89–104.29 pg/mL), galectin 3 presented normal values, human ST2 showed higher than normal values (68.35 with IQR of 34.5–121.1 ng/mL) and IL-1B showed higher values (44.74 with IQR of 42.92–48.49 pg/mL).

We were able to compare the values of FGF-23, human BMP2 and IL 1-beta between our dialyzed cohort and 10 healthy subjects (the same statistical age). The results are presented in Table 4. Dialyzed patients presented higher values of human BMP2 and lower values of FGF-23. There were no statistical differences regarding IL 1-beta values, but the dialyzed patients presented a higher heterogeneity regarding this parameter with a bigger standard deviation.

In order to evaluate the connections between the special markers and cardiac parameters, we performed several multiple regression and logistic regression models (as appropriate) with our special markers as the independent variables. The results are presented in Table 5 and Table 6. IL-1B was positively correlated with EF values and global longitudinal strain and was negatively correlated with left atrium diameter and left ventricle telesystolic diameter. Galectin 3 values were negatively correlated with aortic valve fibrosis and mitral valve calcifications, and human ST2 was negatively correlated with mitral valve calcifications.

None of the specific biomarkers was correlated with mortality during the follow-up.

## 5. Discussion

After more than 50 years since hemodialysis become a reliable method of renal replacement therapy, the first cause of death among HD patients is a cardiovascular-related one. In the face of the CKD and cardiovascular disease continuum, research on better predictors and possible intervention means is mandatory in order to reduce mortality. The aim of this pilot study was to evaluate the correlations between specific biomarkers and echocardiographic parameters in a cohort of HD patients. In addition, we explored the impact on mortality of these specific biomarkers.

The mean age of the cohort was 61.02 years with 56.2% males. Our cohort was younger compared to the ERA-EDTA annual report from 2020 (65.1 years) and similar to several European countries like Finland (61 years), United Kingdom (61 years), Serbia (60.3 years), etc. [33]. In Romania, the mean age of the patients on HD in 2020 was 62.2 years [34]. The management of CKD complications was conducted according to KDIGO guidelines, and as a result, the patients had in-target hemoglobin levels, good nutritional status, reduced inflammation and proper calcium phosphorus product.

Regarding echocardiographic parameters, the EF was within the normal range. The valvular and endomyocardial calcification rate was high as a result of the mineral bone disease evolution. For instance, in a study published by Kraus on 243 HD patients, all of them presented aortic or mitral valve calcifications [35]. Most of the cohort presented left ventricle hypertrophy. This is commonly encountered in dialysis patients, as per previously published data [36]. The cause of this comorbidity among HD patients is multifactorial and is linked besides arterial hypertension and fluid overload to specific CKD complication factors (vitamin D, erythropoietin, calcifications, etc.) [36].

IL-1B is a pro-inflammatory cytokine involved in inflammatory response, cell growth and tissue repair. In patients with myocardial infarction, increased levels of IL-1B are associated with worse cardiac outcomes [17]. Patients on maintenance HD have a pro-inflammatory state with higher IL-1B levels [14,15]. In the face of these statements, we evaluated the correlations between IL-1B and cardiac ultrasound characteristics in this specific population.

The median values of IL-1B in our study was 44.74 pg/mL, being lower than the mean values in a cohort of 390 patients evaluated by Yu (84.82 ± 94.38 pg/mL) [37]. These differences are most likely due to a lower inflammation status in our cohort (CRP 0.43 mg/dL vs./8.46 mg/dL in Yu cohort) [37]. In addition, in another recently published study by Lisowska on 67 HD patients, the mean IL-1B levels were 1.75 pg/mL, much lower than Yu’s results and lower than the values from our cohort [38]. These differences may be in the context of several factors: inflammation and nutrition status, hemoglobin levels and anemia management, dialysis vintage, comorbidities and the number of patients.

Interestingly, the IL-1B values were positively associated with GLS values in our cohort. Each unit increase in IL-1B translated to a 0.011% decrease in the GLS (increase in the absolute value). For instance, in a study that evaluated the effects of Anakinra (interleukin-1 receptor antagonist) in 80 patients with rheumatoid arthritis on coronary and left ventricular function, the patients with higher IL-1B levels presented the highest reduction in GLS after Anakinra administration. In addition, the GLS values in our cohort were higher compared with the ones in the aforementioned study (−17% vs. −33%). One should mention that we did not evaluate the incidence of coronary artery disease in our patients. To our knowledge, no studies have evaluated the relationship between IL-1B and GLS in patients on maintenance HD. Some authors consider that GLS could be a better predictor of left ventricle systolic dysfunction compared to the ejection fraction measurement in patients on dialysis [39].

In our study, IL-1B was positive associated with EF values. Each unit increase in IL-1B generated a 0.022% increase in the EF. No study so far has evaluated the correlation between EF and IL-1B in hemodialysis patients. A nice study by Orn on patients following myocardial infarction evaluated the correlation between IL1-B and the EF values at different time points [40]. Their findings show a negative correlation between EF and IL-1B values, with patients that presented higher IL-1B levels having lower EF values [40]. One should mention that this negative correlation was present at 2 months and 1 year after angiographic intervention, thus not only in acute settings. Our findings seem to be opposite compared with Orn’s cohort, with higher levels of IL-1B being associated with higher levels of EF. In the dialysis settings, there are already some known facts regarding reversed epidemiology. HD patients live longer at higher body mass indexes and, according to some authors, if they present higher cholesterol levels [41,42,43]. These facts are explainable due to the nutritional status, with malnutrition being associated with higher mortality among HD patients [42,43]. Thus, lower body mass indexes and lower cholesterol levels are most likely associated with a malnutrition status in the HD population. Our results are unnatural, as one would expect that higher IL-1B levels would be associated with lower EF. Even though in our cohort, values of IL-1B had no influence on mortality; perhaps patients with higher EF values present higher valvular regurgitation values and the EF is artificially increased (this is just an assumption due to the fact that we did not evaluate the valvular regurgitation status). On the other hand, an overhydration status could lead to an increase in preload and afterload and thus could interfere with our results. We were unable to properly evaluate the hydration status in our cohort, but there was no statistical correlation between ultrafiltration volume and IL-1B levels.

Moreover, IL-1B levels were negatively correlated with left atrium diameter and left ventricle telesystolic diameter. So far, increased left atrium diameter is associated with increased all-cause mortality among the general population [44]. To our knowledge, no studies have evaluated the impact of left atrium diameter on IL-1B levels. Even though left atrium diameter larger than 40 mm is associated with a higher risk of major adverse cardiac events (MACEs) [45], in our study, left atrium diameter did not increase the risk of death.The study of Orn is the only one that evaluated the impact of IL-1B measured after myocardial infarction on left ventricular modeling evaluated through cardiac magnetic resonance at one year after infarction [40]. He showed that increased IL-1B levels at 2 months following myocardial infarction were associated with the left ventricle end systolic index. In our cohort, it seems that the correlation is the opposite from Orn, with respect to the differences regarding cohort size, non-infarct patients in our cohort and different timing in cardiac evaluation [40].

Our results regarding IL-1B and cardiac structure and function seem different from the ones published in the literature, even though no study so far has evaluated this interaction in patients on maintenance HD. The cardiac changes in patients on HD can be different compared to non-HD ones. As we stated earlier, these results could be another reversed epidemiology in the context of dialysis. For instance, we published a study two years ago where we found out that an increase in the EF after one year of maintenance HD increased mortality, even though patients with initial reduced EF presented higher mortality [46]. IL-1B is mainly produced by dendritic cells, tissue macrophages and blood monocytes. In the previously cited study by Orn, one could easily consider that the cause of IL-1B increment was the infarct status. In our cohort, we do not know the cause of IL-1B production. It could be in context of cardiac alterations or could be in the context of another inflammation site. We carefully interpret our results, as cardiac function and structure correlations with IL-1B levels could be a consequence of increased IL-1B, or the heart could be the one that produces IL-1B. One should always keep in mind that the results could be due to chance.

Galectin-3 is a relatively new biomarker that is well studied as a promoter of cardiac inflammation and fibrosis leading to heart failure [19,20]. In hemodialysis-treated patients, higher levels of galectin-3 seem to be associated with increased cardiac and all-cause mortality [21,22,23]. In our study, galectin-3 levels did not influence mortality during the two years of follow-up. One should mention that the study by Liu on 506 HD patients showed that galectin levels above 8.65 ng/mL increased mortality, and the study by Hogas on 88 HD patients found a galectin-3 cut-off value of 23.73 ng/mL as an increased mortality risk [22,23]. The interesting result is that galectin-3 levels are negatively correlated with aortic valve fibrosis and mitral valve calcifications. Previously published papers showed a direct link between galectin-3 and cardiac fibrosis and cardiac remodeling [19,20]. No study from the literature evaluated this relationship among dialysis or non-dialysis patients. It is possible, at first view, to consider these results unnatural. One should expect a patient with proved fibrosis to present higher galectin-3 levels. The valvular disease progresses during the hemodialysis. Patients with increased galectin-3 levels probably are at higher risk of valvular fibrosis and maybe valvular calcification. The interplay between fluid overload, myocardial HD-associated stunning, left ventricle hypertrophy and progression of mineral bone disease is complex, and galectin-3 may play an important role. It is possible that our results could be attributed to chance, or can be explained by the reduced number of patients, but they should not be overlooked. There is definitely an imperative need for future studies to validate our results.

Like galectin-3, sST2 proves its efficiency in predicting MACE, even in CKD patients [24,25]. There are proven links between elevated sST2 levels and cardiac remodeling [26]. The relationship between sST2 and cardiac remodeling is less studied among HD patients, but a meta-analysis on HD patients that gathered more than 1300 patients showed that increased sST2 levels increase all-cause mortality [28]. No study evaluated so far the possible connection with valvular calcifications. Some studies evaluated the imbrication of sST2 and atherosclerotic plaque calcifications. For instance, Luo showed that in patients with non-ST-elevated acute coronary syndrome, higher levels of sST2 were associated with spotty atherosclerotic calcifications, and those with lower levels exhibited large plaque calcifications [47]. In our study, there is a negative connection between sST2 levels and mitral valve calcifications. Previous studies showed that sST2 may suppress the M2 phenotype differentiation of macrophages [48]. M2 macrophages present mostly anti-inflammatory effects and are related to macrocalcification in atherosclerotic plaques. There is a possibility that patients with elevated sST2 present less frequent mitral calcifications following the same pathophysiological mechanisms of atherosclerotic plaques calcification. Vascular calcifications in patients with CKD are evolving more rapidly compared with those in non-CKD patients, most likely in the context of CKD-MBD. In the face of this evolution, one should expect that patients on maintenance HD present a longer period of a pro-atherosclerotic calcification status; thus, they most likely would present the final stage of atherosclerotic plaque evolution, plaque macrocalcification. Considering that mitral valve calcification mechanisms are similar to the ones from large plaque calcifications, our results seem natural and confirm the previous published data, as elevated sST2 levels seem to be correlated with spotty calcifications and not with large ones [47]. These results require further confirmation in prospective studies with larger number of patients in order to reduce bias and/or positive results due to chance.

In our cohort, BMP2 values were not correlated with either cardiac parameter. One should mention that healthy controls presented lower BMP2 levels compared to HD patients. A study by Dalfino showed that patients with chronic kidney disease presented higher levels of BMP2 compared to healthy controls [49]. In addition, he suggested that BMP2 might contribute to vascular calcification due to increased oxidative stress and arterial stiffness [49]. Our results confirm the previous findings.

There are several limitations in our study. First, the small number of patients and the relatively reduced period of follow-up are drawbacks. Due to the small number of cases, the results should be interpreted with caution. One should keep in mind that a drawback of the study is the lack of proper evaluation of hydration status in our cohort. An increase in preload and afterload could influence our results. To our knowledge, this is the first study that explores the interrelation between some specific biomarkers and cardiac remodeling in patients on maintenance hemodialysis. There is an imperative need for future prospective studies with a larger number of patients in order to validate our results.

In conclusion, some of these biomarkers could be used for a better assessment of cardiac function and structure in patients on hemodialysis.

## Figures and Tables

**Table 1 diagnostics-14-00664-t001:** Baseline characteristics.

Parameter	Value *N* = 77 Patients
Age (A + SD) years	61.02 (11.81)
Dialysis vintage (M + IQR) years	4.95 (2.4–7.8)
Sex (male)	45 (56.2%)
Dry weight (A + SD) kg	82.61 (19.52)
BMI (A + SD) kg^2^/m^2^	28.93 (6.46)
Albumin (M + IQR) g/dL *	4.1 (3.9–4.3)
Serum bicarbonate (A + SD) mmol/L	22 (2.27)
Predialysis creatinine (A + SD) mg/dL	8.54 (2.05)
Predialysis urea (A + SD) mg/dL	122.63 (25.9)
Hemoglobin (M + IQR) g/dL	10.85 (10.3–11.6)
C-reactive protein (M + IQR) mg/dL	0.43 (0.26–1.2)
Serum sodium (M + IQR) mmol/L	139 (137.7–140.4)
Serum potassium (A + SD) mmol/L	5.37 (0.64)
Serum calcium (A + SD) mg/dL	8.66 (0.51)
Serum phosphorus (M + IQR) mg/dL	5.39 (4.67–6.3)
Calcium phosphorus product (A + SD) mg^2^/dL^2^	47.86 (13.06)
Parathyroid hormone (M + IQR) ng/mL	428 (197–713.5)
Thrombocytes (M + IQR) N/mm^3^	219,500 (174,000–254,000)
eKT/V (A + SD)	1.6 (0.2)
Mean ultrafiltration volume (A + SD) mL	2343 (587)
Deaths N (%)	17 (22.1%)

Legend: A = average, SD = standard deviation, M = median, IQR = interquartile range, kg = kilogram, m = meter, g = grams, dL = deciliter, mmol = millimole, L = liter, mg = milligram, ng = nanogram, mm = millimeter, N = number, * 3 patients presented values of serum albumin lower than 3.5 g/dL.

**Table 2 diagnostics-14-00664-t002:** Echocardiography parameters.

Parameter	Value
Left atrium diameter (A + SD) mm	41.87 (4.7)
Right ventricle diameter (M + IQR) mm	27 (26–28)
Aortic atheromathosis N (%)	60 (76.9%)
Endomyocardial calcifications N (%)	55 (70.5%)
Kinetics dysfunction N (%)	33 (42.3%)
Aortic valve calcifications N (%)	53 (67.9%)
Aortic valve fibrosis N (%)	63 (80.8%)
Mitral valve calcification N (%)	61 (78.2%)
Mitral valve fibrosis N (%)	61 (78.2%)
Let ventricle telediastolic diameter (M + IQR) mm	53 (49–58)
Left ventricle telesystolic diameter (A + SD) mm	38.78 (6.51)
Waves E/A rapport (M + IQR)	0.7 (0.4–1)
Ejection fraction (M + IQR) %	51 (43–54)
Global longitudinal strain (M + IQR) % (absolute value)	14 (12–17)
Interventricular septum (M + IQR) mm	13 (12–14)
Left ventricle mass (M + IQR) g	254.5 (203–304)

Legend: A = average, SD = standard deviation, M = median, IQR = interquartile range, N = number, mm = millimeter, g = gram.

**Table 3 diagnostics-14-00664-t003:** Special parameters analysis.

Parameter	Value
FGF-23 (M+ IQR) pg/mL	47.93 (21.89–104.29)
Galectin 3 (M+ IQR) ng/mL	3.9 (2.62–9.62)
Human ST2 (M+ IQR) ng/mL	68.35 (34.5–121.1)
Human BMP2 (M+ IQR) pg/mL	712.7 (451–725.9)
IL-1B (M + IQR) pg/mL	44.74 (42.92–48.49)

Legend: FGF = fibroblastic growth factor, ST = suppression of tumorigenicity, BMP = bone morphogenetic protein, IL-1B = interleukin 1Beta, M = median, IQR = interquartile range, pg = picogram, mL = milliliter, ng = nanogram.

**Table 4 diagnostics-14-00664-t004:** Comparison between the dialyzed cohort and healthy subjects for FGF-23, human BMP2 and Il 1-beta.

Parameter	Dialyzed Patients	Healthy Controls	*p* Value
FGF-23 (A + SD) pg/mL	75.98 (76.97)	290.31 (173.91)	<0.001
Human BMP2 pg/mL	591.18 (341.69)	242.11 (22.82)	0.0061
IL 1-beta	65.14 (82.53)	46.03 (2.84)	0.468
Age (years)	61.02 (11.81)	59.12 (7.86)	0.587

Legend: FGF = fibroblastic growth factor, BMP = bone morphogenetic protein, IL-1B = interleukin 1-beta, A = average, SD = standard deviation, pg = picogram, mL = milliliter.

**Table 5 diagnostics-14-00664-t005:** Multiple regression models with specific biomarkers as independent variables.

Parameter	EF	GLS (Absolute Value)	LA	LVTSD
IL-1B pg/mL	Coef = 0.022, *p* < 0.0001	Coef = 0.011, *p* = 0.003	Coef = −0.024, *p* = 0.031	Coef = −0.021, *p* < 0.0001
Galectin 3 ng/mL	-	-	-	-
Human ST2 ng/mL	-	-	-	-
Human BMP2 pg/mL	-	-	-	-
FGF-23 pg/mL	-	-	-	-
R square adjusted	0.689	0.109	0.054	0.73

Legend: FGF = fibroblastic growth factor, ST = suppression of tumorigenicity, BMP = bone morphogenetic protein, IL-1B = interleukin 1-beta, M = median, IQR = interquartile range, pg = picogram, mL = milliliter, ng = nanogram.

**Table 6 diagnostics-14-00664-t006:** Logistic regression models with specific biomarkers as independent variables.

Parameter	Aortic Valve Fibrosis	Mitral Valve Calcifications
IL-1B pg/mL	-	0.98 (0.97–1.00), *p* = 0.071
Galectin 3 ng/mL	0.92 (0.85–0.99), *p* = 0.042	0.88 (0.79–0.98), *p* = 0.025
Human ST2 ng/mL	0.99 (0.98–1.00), *p* = 0.084	0.98 (0.97–0.99), *p* = 0.01
Human BMP2 pg/mL	–	–
FGF-23 pg/mL	–	1.01 (0.99–1.02), *p* = 0.085
Nagelkerke R square	0.154	0.37
AUC (95%CI)	0.655 (0.531–0.765)	0.809 (0.697–0.894)

Legend: FGF = fibroblastic growth factor, ST = suppression of tumorigenicity, BMP = bone morphogenetic protein, IL-1B = interleukin 1-beta, M = median, IQR = interquartile range, pg = picogram, mL = milliliter, ng = nanogram.

## Data Availability

Anonymized data will be available after request to the corresponding author—Adelina Mihaescu, e-mail: mihaescu.adelina@umft.ro.

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
