# Peer review of "Novel Biomarkers in Evaluating Cardiac Function in Patients on Hemodialysis—A Pilot Prospective Observational Cohort Study"

_diagnostics, 2024, doi:10.3390/diagnostics14060664_

Round 1

Reviewer 1 Report

Comments and Suggestions for Authors

This is a prospective observational cohort study on 77 patients of whom the plasma levels of Interleukin 1-Beta, Galectin 3, human suppression of tumorigenicity factor 2, bone morphogenetic protein 2 and fibroblastic growth factor 23 were measured when selected. The results showed positive correlation between Interleukin 1-Beta with ejection fraction and global longitudinal strain and negative correlation with left atrium diameter and left ventricle systolic diameter. Galectin 3 values negatively correlated with aortic valve fibrosis and mitral valve calcifications and human suppression tumorigenity factor 2 negatively correlated with mitral valve calcifications. 

The methodology was great. The discussion was great. These are new data for clinicians and researchers. The English language was great, except a few minor typographic errors.   

Author Response

Thank you for your positive feedback. We reread the article and corrected all the identified typos. Thank you again!

Reviewer 2 Report

Comments and Suggestions for Authors

Patients on programmed hemodialysis (PGD) are known to develop both functional and structural changes in the cardiovascular system. However, in such patients, “typical” symptoms lose their value, which justifies the need to use additional diagnostic methods that allow differentiating signs of heart failure and symptoms of overhydration characteristic of PGD. Such methods, in addition to traditional clinical assessment, include molecular biomarker analysis and echocardiography. This makes the implementation of this study timely and relevant.

Weaknesses of the study:

1) I think it would be appropriate to add information regarding the number of patients with chronic overhydration due to inadequately performed PGD, since these patients, who have increased preload, the dynamics of biomarkers and prognosis with a high degree of probability may depend on this factor. It is necessary to analyze the level of biomarkers depending on the fact of increased preload and afterload. It is possible that this may help explain the paradoxical correlations found in this study, as well as the lack of association between biomarker levels, particularly galectin and St2, and mortality;

2) I consider it necessary to supplement the data regarding the presence of hypoalbuminemia, indicating the number of such patients;

3) the fact that patients with hypertension were included in the study for the first time becomes known only in the “discussion” section; this information should be supplemented with the “materials and methods” section; the explanation for the discrepancy between our own results and the data of other researchers is unconvincing (direct correlation of IL-1B and VWF), and noteworthy is the fact that according to this study, some BM behave typically, while others behave atypically. In particular, there is an inverse correlation between IL-1β and LVEF, which contradicts the well-known data that IL-1β reduces the beta-adrenergic response of L-type calcium channels through a cyclic adenosine monophosphate-independent mechanism and increases the expression of nitric oxide synthase (NOS). ) in cardiomyocytes, which leads to an increase in the activity of nitric oxide (NO) and a decrease in myocardial contractility; small sample of patients.

Author Response

First, we would like to thank you for the suggestions. We appreciate your comments and we hope that the implementation of your suggestions will improve the article.

From the beginning, we agree with you regarding the inverse epidemiology in patients on HD. In addition, as you stated in the introduction of your comments, typical is not typical anymore in these kind of patients, in the face of cardiovascular disease symptoms.

Comment 1:

We understand your concern regarding the overhydration status and its possible interaction with outcomes or the possible inference with the results. Some patients tend to become overhydrated in HD due to several reasons: inefficient ultrafiltration target due to hypotension, arrhythmias, angina, cramps, and so on. One should keep in mind that some patients do not respect the daily fluid income and present to the dialysis session overhydrated. On the other hand, we specific chose the day of the evaluation after the longest period without dialysis in order to better this issue. The overhydration status is difficult to assess clinically or using only the weights values in dynamic. One of the practical ways to evaluate the hydration status of a HD patient would be the bio impedance. In this dialysis patients, the bio impedance evaluation of the hydration status is possible only once a year. In 2021, the bio impedance was performed in April and we considered that is inappropriate to use those data for hydration status. In addition, current literature suggests that an ultrafiltration rate of 12.5 ml/kg/hour should be the upper limit regarding ultrafiltration. Values higher than this are associated with worse outcomes. Only 2 of the patients from this cohort required ultrafiltration volumes higher than this value (one with 13.75ml/kg/hour and one with 14.5 ml/kg/hour- data not presented in the article). On the other hand, after re-performing the statistical part, we did not found any correlations between any of biomarkers and ultrafiltration expressed as ml/kg/hour. We added in the revised version in the limitations of the study the lack of the real hydration status in our cohort.

Comment 2:

We added in the baseline characteristics table – table 1 – the fact that only 3 patients presented values of serum albumin lower than 3.5 g/dl.

Comment 3:

We added in the material and methods the recording of arterial hypertension presence and in the results we specified the percentage of hypertensive patients (92.2% representing 71 out of 77 patients). We agree with your comments. Indeed some biomarkers presented expected results, while other presented paradoxical ones. As you correctly present the physiopathological path of IL-1B and the ejection fraction. We are aware of the limitations regarding the correct assessment of cardiac function in our cohort. As we mentioned in the discussions, Orn provided important insights regarding cardiac function and structure and its relationship with IL-1B.Unlike us, he indeed used the cardiac magnetic resonance evaluation. We know and clearly specified in the discussion section that our result should be carefully interpreted due to the reduced number of patients. Statistically, in a small cohort, the risk of bias and results due to chance increase. On the other hand, our study could be a starting point for studies with larger cohorts and perhaps using cardiac evaluations better than echocardiography.

In the end, we would like to thank you again for the constructive suggestions and hope that our responses satisfy you.

Reviewer 3 Report

Comments and Suggestions for Authors

-        English proof editing for some minor corrections is recommended.

-        The abbreviations need to be explained when first introduced (e.g. HD), and should be used consistently thereafter.

-        The aim of the study needs to be written more precisely. Namely, the following statement belongs to the Methods section: „ ... we conducted a prospective observational cohort study between October 89 2021 and October 2023 where we evaluated…”.

-        The following part of the text in the Methods section is repeated after each parameter: “The samples were centrifuged for 15 minutes at 1000xG at a temperature raging from 2 to 8 degrees Celsius in the first 30 minutes after collection. The supernatant was collected for the assay”. Instead of repeating, one joint statement for all parameters should be written.

-        Instead of “creatine” it should be “creatinine” in the Results section.

-        The limitations of the study should be mentioned before the conclusions.

Comments on the Quality of English Language

English proof editing for some minor corrections is recommended.

Author Response

We would like the reviewer for its time in evaluating our article and for the suggestions. We hope that implementing the suggested changes will increase the quality of the manuscript. We will give a point by point response.

  1. We proof edited the revised form of the manuscript.
  2. We know specify the abbreviations when first introduced.
  3. We rewrote the aim of the study and moved the statement in the methods section.
  4. We performed the required change.
  5. We performed the required change
  6. We moved the limitations before conclusions.

Round 2

Reviewer 2 Report

Comments and Suggestions for Authors

I recommend supplementing the “discussion” section with a more detailed discussion of the possible mechanisms of atypical changes in biomarkers

Author Response

Reviewer 2 comment:

I recommend supplementing the “discussion” section with a more detailed discussion of the possible mechanisms of atypical changes in biomarkers.

Response:

Thank you for this suggestion. We detailed in the discussion section the possible explanation for our results. We marked them in green this time. He hope that this changes will satisfy your request.